# How the Intensity of Night Shift Work Affects Breast Cancer Risk

**DOI:** 10.3390/ijerph18094570

**Published:** 2021-04-26

**Authors:** Marta Szkiela, Ewa Kusideł, Teresa Makowiec-Dąbrowska, Dorota Kaleta

**Affiliations:** 1Department of Hygiene and Epidemiology, Faculty of Health Sciences, Medical University of Lodz, 90-752 Lodz, Poland; dorota.kaleta@umed.lodz.pl; 2Department of Hygiene and Health Promotion, Faculty of Health Sciences, Medical University of Lodz, 90-752 Lodz, Poland; 3Department of Spatial Econometrics, Faculty of Economics and Sociology, University of Lodz, 90-255 Lodz, Poland; ewa.kusidel@uni.lodz.pl; 4Nofer Institute of Occupational Medicine in Łódź, Department of Environmental and Occupational Health Hazards, 91-348 Lodz, Poland; tmd@imp.lodz.pl

**Keywords:** shift work, night work, breast cancer, risk factor, light at night, occupational cancer, working condition

## Abstract

Background—In 2019, the IARC concluded that “night shift work is probably carcinogenic to humans (Group 2A), based on limited evidence from human epidemiological studies and sufficient evidence of cancer and strong mechanistic evidence in experimental Animals.” The negative health consequences of night shift work may depend on how the night shifts are scheduled. The aim of this study was to investigate how the characteristics of night work affect the risk of developing breast cancer. Methods—A case–control study was conducted in 2015–2019 in the Lodz region. The case group included 494 women with breast cancer, while the control group included 515 healthy women. Results—Night work was found to be the third most important factor regarding breast cancer after a high BMI and a short or no breastfeeding period and before factors such as early menstruation, late menopause, no pregnancy, and smoking. The harmful effects of night work were influenced by its intensity, frequency, rotation, and the number of night shift years worked. Night work increases the breast cancer risk by 2.34 times, and high-intensity night work increases the breast cancer risk by 2.66 times. Conclusions—Appropriate ergonomic recommendations for night shift work for employers should be considered.

## 1. Introduction

Working at night is inconsistent with the biological rhythm of human life. The body adapts to its circadian rhythm thanks to light stimuli. The master biological clock, which is located in the suprachiasmatic nuclei (SCN) (located in the dorsal brain, dorsal from the optic junction and latero-abdominal from the third ventricle of the brain), helps the body get used to 24 h environmental cycles. The cells of the biological clock, which are present in all organs of the human body, have the ability to regulate endogenous biological rhythms [1,2]. Working at night and exposure to light at night disturb the circadian rhythms. Among the effects of a circadian rhythm disturbance on the endocrine system is a change in the rhythm of the secretion of many hormones, including prolactin, glucocorticoids, adrenocorticotropic hormone, corticoliberin, serotonin, and melatonin [3,4]. Melatonin production is always highest at night and low melatonin levels occur during the day. Melatonin acts as a coordinator of biological rhythms by providing the body with information about the time of the day and the expected time of the year [5]. Exposure to light at night among shift workers can range from 50 to 100 photopic lux at the eye and some exposures may exceed 200 lux [6]. Shift work also includes changes in the timing of daily activities, such as eating, sleeping, physical activity, and social patterns. Smoking tobacco and alcohol consumption may also differ in shift workers [7].

In 2007, the International Agency for Research on Cancer (IARC) concluded that “shift work is probably carcinogenic to humans (Group 2A), based on limited evidence of carcinogenicity for breast cancer from eight human studies, in addition to sufficient evidence from studies in experimental animals” [8]. In 2019, the IARC concluded that “night shift work is probably carcinogenic to humans (Group 2A), based on limited evidence from human epidemiological studies and sufficient evidence of cancer and strong mechanistic evidence in experimental Animals” [9]. The first hypothesis that exposure to light at night might increase the risk of breast cancer appeared in 1987 [10]. Shift work may be considered the most important occupational cause of breast cancer. The results of epidemiological studies conducted in the United Kingdom [11] and the USA [12] based on the PAF (population attributable fraction) method (proportion of disease that could be prevented by decreasing population exposure to modifiable risk factors) indicated that approximately 4.5% (U.K.) to 5.7% (USA) of breast cancers could be due to shift work (including night work).

The negative health consequences of night work may depend on how the night shifts are scheduled. Most epidemiological studies have found that higher night work intensity (number hours or night shifts per day/week/month) is associated with a higher risk of breast cancer [13,14,15,16]. The next characteristic of night shift work that may affect breast cancer risk may be the number of consecutive night shifts. According to previous epidemiological studies, night work lasting ≥3 consecutive nights may increase the risk of breast cancer [14,15,16,17,18,19]. Another characteristic of night shift work that may affect breast cancer risk may be the direction of rotation. In night shift work, there is forward (the night shift follows the afternoon shift and the afternoon shift follows the day shift) and backward (the day shift follows the afternoon shift and the afternoon shift follows the night shift) rotation. Forward rotating allows for sufficient time for recovery between the shifts [20]. Chronotype (individual variations of sleep/wake times, influenced by environmental light, genetics, and human development stages) may also modify potential health risks in shift workers [21]. People with a morning (early risers) chronotype go to bed early and wake up early. People with an evening (night owl) chronotype go to sleep late and get up late, causing a phase delay of their circadian system. People with an evening chronotype have a higher risk of several negative shift work health consequences, e.g., higher blood pressure, increased resting heart rate, sleep apnea, lower HDL levels, higher BMI, and type 2 diabetes [22].

Breast cancer is the most frequently diagnosed cancer among women (24.2% of all cancers). It is also a leading cause of cancer-related deaths in women (15.0%) [23]. In Poland in 2017, breast cancer was the most common cancer among women (22.5% of all malignancies) and the second (after lung cancer) cause of death due to malignant tumors (14.8% of all malignant neoplasms) [24]. The most important risk factors for breast cancer appear to be older age, the carrier of mutations in some genes (primarily BRCA1 and BRCA2), the occurrence of breast cancer in relatives, early age of first menstruation, late menopause, late age of first delivery, long-term hormone replacement therapy (HRT), exposure to ionizing radiation, some mild breast proliferative diseases, a low level of physical activity, improper diet, obesity, alcohol, smoking, and night shift work [25,26].

Since in our previous article [27], we could confirm the significant impact of night work on breast cancer, the aim of this study was to investigate how the intensity, frequency, and type of night work rotation (forward, backward) affected the risk of breast cancer.

## 2. Materials and Methods

### 2.1. Design and Study Population

The case–control study was conducted in 2015–2019 in the Lodz region. The case group included 494 women over 35 years old, who had been diagnosed with malignant breast cancer and had a tumor resection or mastectomy. The control group included 515 women without breast cancer. The women from the case group were patients of the Oncological Surgery Department and the Second Department of Oncological Surgery, Oncological Surgery Clinics of the Provincial Specialist Hospital M. Kopernik in Lodz; the surgery department of Poddębice Health Center; the Provincial Specialist Hospital M. Skłodowska-Curie in Zgierz. The anthropometric data of the research participants were assessed using a self-assessment questionnaire. A detailed description of the study design and study population, including the selection of women for the case/control groups and a description of the research tool, can be found in our previous article: “Night Shift Work—A Risk Factor for Breast Cancer” [27]. The study design received a positive opinion from the Bioethics Committee at the Medical University of Lodz (RNN/236/15/EC of 22 September 2015).

### 2.2. Measures

#### 2.2.1. Outcome Variables

The outcome variable was breast cancer. The criteria for including women in the case group were histopathologically confirmed breast cancer or/and mastectomy and no history of other cancers. The criterion for including women in the control group was no history of breast cancer.

#### 2.2.2. Explanatory Variables

##### Shift Work

Respondents were asked whether they ever worked for at least 6 months. We were interested in all types of employment: full-time or part-time work, self-employment, work in a family business, and work in the army or on a farm. If respondents ever worked for at least 6 months, they were asked if they ever worked a night shift, with the possible answer options being “Yes” and “No.” Respondents working night shifts were asked whether they had worked three nights or less on a specific shift or more than three consecutive night shifts, and then they were transferred to the next shift. We also asked about the order of the shifts. If the order of shifts was I (morning)→II (afternoon)→III (night), we named it the “forward rotation.” If the order of shifts was III→II→I, we named it the “backward rotation.” The respondents were also asked how many times a month they worked at night. Finally, we also asked about the starting point for shift work, with possible answers being “Within 12 months before breast cancer,” “1–5 years before breast cancer,” “5–10 years before breast cancer,” “Over 10 years before breast cancer.” Finally, from the group of night shift employees, we derived four variables measuring the intensity of such work:3 days or less/more than 3 days of consecutive night shifts (one after another);Forward/backward rotation of shift work;Less/more than 10 years of night work before illness developed;Number of night work years.

##### Sociodemographic Data

The respondents reported their birth date (we analyzed age in years using five variables (≤47, 48–58, 59–69, 70–80, >80)), place of living (countryside, small-sized town (<50,000 inhabitants), medium-sized town (51,000–100,000 inhabitants), large-sized town (>100,000 inhabitants)), marital status (married, widow, never married, divorced, separated), and education level (ISCED 5–6: higher education, ISCED 4: post-secondary education, ISCED 3+: secondary education, ISCED 3−: vocational education, ISCED 1: primary education).

##### Other Breast Cancer Risk Factors

Breast cancer risk factors that were assessed included the age of first menstrual period and age of menopause, pregnancy history (number of pregnancies, age of first delivery, and duration of breastfeeding), smoking status, and body mass index.

#### 2.2.3. Statistical Methods

We use contingency tables and multiple logistic (logit) regression to estimate the crude (Table 1 and Table 2) and adjusted (Table 3) odds ratios (ORs). Statistical calculations were carried out in (1) a Microsoft Excel 365 spreadsheet (Microsoft, Redmond, Washington, USA) (frequency distribution of the variables and contingency tables), (2) Gretl ver. 2020 (open-source statistical package) (logit model and backward elimination procedure), and (3) Statistica 13.3 (StatSoft, Tulsa, OK, USA). As for the results in Table 3, each column represents a separate logit regression for which predictor reduction in the logit model was carried out using a backward elimination model selection procedure, which started with the most general model and eliminated one variable at a time until the best model was reached (i.e., when all the right-side variables were statistically significant for *p* < 0.05). Exponentiating the estimators from Gretl package produced the odds and the direct estimators from logistic regression in Statistica. Each of the A–G columns of Table 3 shows statistically significant ORs from separate logit regressions with the same set of the confounding factors: BMI > 25, breastfeeding < 6 months, menstruation age: 10–12, menopause age: 55+, living in the countryside, widow, smoking, and no pregnancies (for each variable, the reference group contained all other respondents who did not meet the given condition), as well as different night work intensity: (A) any night shift work, (B) working nights more than ten years before illness, (C) forward rotation of night work, (D) more than three consecutive night shifts, (E) more than 10 years of working nights, (F) trend of years of working nights, and (G) more than 10 years of working nights and more than three consecutive night shifts. The purpose of the analysis from Table 3 was to show how the intensity of shift work, defined using A–G, affected disease with the control of the same set of confounding factors.

## 3. Results

Considering the aim of the study (an assessment of how the intensity, frequency, and type of shift work rotation, especially night work, affect the risk of breast cancer), the analysis covered 973 women who answered the questions regarding night shift work (out of 1009 women). Among these women, 478 patients had been diagnosed with breast cancer (case group) and 495 women without a breast cancer diagnosis (control group). As we showed in our previous study [27], the case and control groups differed in nearly all variables. Most of these differences were due to well-documented risk factors of breast cancer: the breast cancer risk increases with age, high BMI, young age of the first menstruation, late age of menopause, a multiplicity of pregnancies, short period of breastfeeding (or lack of breastfeeding), and smoking. Social factors significantly differentiated the groups: more breast cancer patients live in the village, were widows, and had a low education (ISCED 3−, ISCED 1) level. For women working night shifts, the OR was 2.61 (95% CI = 1.94–3.53), which meant that the risk of breast cancer was 161% higher for women working night shifts than for women not working night shifts.

In Table 1, we examined a set of variables similar to those in [27], but with a particular focus on night shift workers. The results show that the night shift workers had a higher risk of breast cancer among the considered subgroups of risk factors for breast cancer. This is shown in Table 1, in which all statistically significant ORs (in bold) indicate that the risk of breast cancer for night shift workers was usually 2–3 times higher than in the group of employees not working shifts. This means that the harmfulness of night work (in terms of increasing the risk of cancer) was not diminished by a young age, low BMI, late age of the first menstruation, young age of menopause, lack of pregnancies, or a long period of breastfeeding.

Table 2 shows the diversity of night workers in terms of the “intensity” of this type of work. For all these results, the reference result was OR = 2.61, which was obtained as the total number of women who worked shifts to women who did not work shifts. The results in Table 2 confirmed that the “intensity” of night work was of significant importance for the risk of breast cancer. As for the length of night shift work, women who worked this way consecutively for more than 3 days had a 202% higher risk of breast cancer than those who did not work shifts (OR = 3.02), while workers who worked less than three consecutive days on a night shift had only a 102% higher risk of breast cancer (OR = 2.02). In other words, the breast cancer risk for people who had not worked for more than three consecutive night shifts was lower than for those who had worked nights for more than three consecutive nights. Most of the night workers had a third shift following the afternoon shift (forward rotation) and these women had a lower risk of breast cancer than the women who had an afternoon shift after the night shift (OR = 2.58 vs. OR = 3.31). In other words, the breast cancer risk for women who worked nights after the afternoon shift was lower than for women who stayed on the afternoon shift after the night shift. Taking into account the working time before the onset of the disease, when it exceeded 10 years, the risk of breast cancer was 191% higher than for women who did not work shifts (OR = 2.91). This result was higher than for the total number of employees working in the third shift (OR = 2.61) and concerned the majority of women working on night shifts (only 8 people out of 168 patients working on night shifts declared that they had developed breast cancer less than 10 years after starting to work night shifts). The intensity of night work versus the intensity of non-night work shows that if the night work lasted no more than 9 years, it was not significant for the risk of breast cancer. Only working for a time over 10 h significantly differentiated shift workers and non-shift workers, with the maximum for the group of 20–29 h shift workers, for whom OR = 3.16, which meant that people working shifts for 20–29 h had a 216% higher risk of breast cancer than people not working in the night. Interestingly, OR was smaller for the higher age categories, which was related to the small sample size, but may also reflect the fact that for the oldest people, age may overcame all the other risk factors.

Finally, we estimated logit models, which initially included all ten disease predictors listed in Table 1 plus one of the shift work variables from Table 2. Table 3 shows the statistically significant OR estimators for the seven versions of the logit model with the differently defined night work “intensity” (listed in the footnote to the last row); also see Section 2.2.3.

The estimators from the first (OR) column of Table 3 confirmed that the significant risk factors for cancer were high BMI (increased the odds of illness by 3.43 times), short breastfeeding time (increased by 1.74 times), early age of menstruation (increased by 1.15 times), late menopause (+86%), living in the countryside (+76%), widow status (+62%), smoking cigarettes (+57%), and no pregnancies (which decreased the odds by 45%). Night work increased the odds by 1.24 times (the adjusted OR for night shift work was lower than the crude one from Table 2 since we took into account many other cancer factors).

From the last row of Table 3, we can conclude that working nights more than ten years before illness (B) had nearly the same impact as having worked a night at all (OR = 2.21 vs. 2.24). Forward rotation (C) had a little lower impact than having worked a night at all. If the number of shift work years was greater than 10 (E), the odds of breast cancer increased to OR = 2.40, similar to working in the system for more than three consecutive nights (D), for which OR = 2.44. The trend coefficient for the number of night shift working years (F) was statistically significant (*p* < 0.008) and meant that every additional year of night shifts increased the risk by 3.1%. Finally, if we combined the (D) and (E) factors and selected people who worked nights for more than ten years and more than three consecutive nights, the OR increased to 2.66 and meant that such people have odds of developing breast cancer that was 2.66 times higher than all the others.

## 4. Discussion

Some epidemiological studies found that a longer duration (number of years, number of shifts) of rotating and night shift work increases breast cancer risk [6,13,14,15,16,28]. In our study, we found that more consecutive night shifts and more years of night shifts increased the risk of breast cancer.

In our study, we found that every additional year of night shifts increased the risk by 3.1% with Ptrend = 0.008. A study by Jones et al. found no significantly increased risks in relation to hours worked per night (Ptrend = 0.62), nights per week on night shift (Ptrend = 0.066), cumulative years of employment as a night shift worker (Ptrend = 0.51), and cumulative hours of night shift work (Ptrend = 0.51). There was a significant positive trend with average hours per week, adjusted for age only (Ptrend = 0.038) or fully adjusted (Ptrend = 0.035). There were no significantly increased risks with being a night shift worker in the last 10 years by type of occupation. There were no significant associations with the age the participant started night shift work (Ptrend = 0.89), whether night shift work started before (*p* = 0.73) or after (*p* = 0.90) first pregnancy, or by time since they last worked night shifts (Ptrend = 0.38) [26]. In a study by Pahwa et al., the researchers found that almost half (45%) of the women who developed breast cancer worked in health care (particularly in nursing) and social assistance, then in the food services sector (18%), trades (11%), and manufacturing (11%). The author stated that an estimated 2.0 to 5.2% of newly diagnosed breast cancer cases were probably attributable to shift work involving nights [15]. In the study by Cordina-Duverger et al., the researchers pooled data of five population-based case–control studies from Australia, Canada, France, Germany, and Spain. It was observed that among premenopausal women, the breast cancer odds ratio was 1.26 (1.06–1.51). For night shifts ≥10 h long, the breast cancer odds ratio was 1.36 (1.07–1.74). For working ≥3 nights/week, it was 1.80 (1.20–2.71), and for both duration of night work ≥10 years and exposure intensity of ≥3 nights/week, the breast cancer odds ratio was 2.55 (1.03–6.30), which were very similar results to ours. It was found that the breast cancer odds ratio was higher in current or recent night workers (OR = 1.41 (1.06–1.88)) than in those who had stopped night work more than 2 years ago [16]. In our study, we found that women who started shift work more than 10 years before diagnosis faced almost twice as much risk of breast cancer as non-shift workers. Our results show that the risk was even greater since the OR for women working nights was 2.91. Wegrzyn et al. compared two cohorts, namely, NHS and NHS II, and the authors observed that women with the highest levels of shift work (≥30 years in the NHS, ≥20 years in the NHS2) had more additional risk factors for breast cancer compared with those who had never done shift work. Women with the highest levels of shift work were more often overweight and obese, more likely to have had menarche before age 12 years, and more likely to be current smokers (with more pack-years of smoking). In our study, we took into account the additional risk factors and the results showed a lower, but still significant, impact of night shift work. Regarding the NHS, the authors observed no association between breast cancer risk and the duration of rotating night shift work (for ≥30 years vs. 0 years: HR = 0.95, 95% CI = 0.77–1.17; *p* for trend = 0.63). While in the NHS2, 20 years or more of rotating night shift work was associated with a significantly higher breast cancer risk (for ≥20 years at baseline vs. 0 years: HR = 2.15, 95% CI = 1.23–3.73; *p* for trend = 0.23). Women with 20 years or more of cumulative rotating night shift work exposure had a marginally significant higher risk of breast cancer (for ≥20 years of cumulative shift work vs. 0 years: HR = 1.40, 95% CI = 1.00–1.97; *p* for trend = 0.74) [29]. In the prospective Sister Study cohort analysis, Sweeney et al. found little to no increase in breast cancer risk associated with work schedule characteristics (ever working rotating shifts: HR = 1.04, 95% CI = 0.91–1.20; ever working rotating night shifts: HR = 1.08, 95% CI = 0.92–1.27; ever working at night: HR = 1.01, 95% CI = 0.94–1.10; ever working irregular hours: HR = 0.98, 95% CI = 0.91–1.06). They did observe an association between rotating shift work at night (>0 to 5 years vs. never: HR = 1.30; 95% CI = 1.05–1.61), short-term night work (>0 to 5 years vs. never: HR = 1.12; 95% CI = 1.00–1.26), and an increased risk of breast cancer [30]. Engel et al. analyzed the relationship between occupation and female breast cancer in the literature from 2002 to 2017. They observed that occupational exposures to night shift work, some chemicals, ionizing radiation, sedentary work, and job stress may increase the risk of breast cancer. These authors also observed that women working in medical professions, as scientific-technical staff, as flight attendants, and in some production positions, sales, and retail (that is wherever shift work occurs) are likely to have a higher breast cancer risk [31]. In a Norwegian case–control study within a cohort of 49,402 Norwegian nurses, Lie et al. found that there was a significantly higher breast cancer risk for women who worked ≥5 years with ≥6 consecutive night shifts (OR = 1.8, 95% CI = 1.1–2.8) [18]. In a Norwegian case–control study, Lie et al. found a relation between cumulative years working in night shifts and an increased risk of breast cancer. Statistically significant associations were observed between breast cancer and work durations of ≥5 years with ≥6 consecutive night shifts. Increased breast cancer risk was found in those working permanent night shifts and long-term day–night rotating shifts. These authors observed that the long duration of shift work throughout the years is also related to the development of estrogen- and progesterone-positive tumors, especially among young women with intensive (12 h) shifts [17]. In a cross-sectional study on nurses, Gómez-Salgado et al. found that higher breast cancer risk was found for nurses who had been working night shifts for more than 16 years, ≥500 night shifts, and those who had been regularly working ≥3 nights per month for >10 years [32].

Schernhammer et al. analyzed the relationship between breast cancer risk and working night shifts in a Nurses’ Health Study. The authors observed a higher breast cancer risk among women who worked night shifts for 1–14 years (RR = 1.08, 95% CI = 0.99–1.18) or 15–29 years (RR = 1.08, 95% CI = 0.90–1.30) [33]. In a cohort study among premenopausal nurses that involved 12 years of follow up, Schernhammer et al. found that working night shifts for more than 20 years increased the relative risk of breast cancer (compared to women never working night shifts) (RR = 1.79, 95% CI = 1.06–3.01) [34].

An important factor related to night work is the chronotype. We did not include it in our study, which can be considered a limitation of the study. In the pilot study, we observed that the answers to the chronotype questions were too distant in time and posed too much difficulty for the respondents; therefore, we found them to be unreliable. In a case–control study of night shift work and breast cancer risk among women in the Danish military, Hansen and Lassen found that women who had ≥3 consecutive night shifts for ≥6 years and women with a morning chronotype preference had a higher breast cancer risk (OR = 3.9, 95% CI = 1.6–9.5) [35]. In a cross-sectional study, Pepłońska et al. asked about the chronotypes “lark” (morning chronotype) or “owl” (evening chronotype). The authors found that prolonged (>15 years) night shift work could be associated with increased estradiol levels among morning chronotype postmenopausal women [36]. In the California Teachers Study (CTS), Hurley et al. observed that those with a definite evening chronotype had an increased breast cancer risk (crude OR = 1.24, 95% CI = 1.10–1.40; fully-adjusted models OR = 1.20, 95% CI = 1.06–1.35) compared to those with a definite morning chronotype [37].

The strengths of the study were an extensive questionnaire and a large study group. The limitations of the study were the self-assessment questionnaire, the lack of reliable data on the relationship between chronotype and shift work tolerance, and that some of the questions concerned distant events. Another limitation was that the anthropometric indicators were evaluated using a self-assessment method.

## 5. Conclusions

The negative impact of shift work, especially night work, on women’s health is unquestionable. In this study, we focused on the impact of night work and its intensity on breast cancer risk. The breast cancer risk factor for night-shift workers can be almost 4 times higher than for women who do not work nights. Night work is the third most crucial breast cancer risk factor (after a high BMI and a short or no breastfeeding period). However, the harmful effects of shift work are influenced by its intensity, frequency, and rotation. The forward rotation had a little lower impact than working nights, whereas more than 10 years of night shift or working in the system for more than three consecutive nights increased the risk of breast cancer. Since night work and its intensity is a modifiable risk factor, if women’s night work cannot be ruled out, it should be planned to be as harmless as possible. This means that night shifts should not occur one after another for more than three nights, a more favorable type of shift rotation is the forward one (night shift occurs after the afternoon one), and the years of night work should be as short as possible.

## Figures and Tables

**Table 1 ijerph-18-04570-t001:** The crude odds ratios of breast cancer in night workers compared to non-night workers in each category.

	Case			Control					
Categories	Night Workers	Non-Night Workers	Percentage of Night Workers	Night Workers	Non-Night Workers	Percentage of Night Workers	OR	95% CI	
Sum	168	310	35.1%	85	410	17.2%	2.61	1.94	**3.53**
Age									
≤49 years	14	56	20.0%	10	85	10.5%	2.13	0.88	5.12
50–59 years	47	64	42.3%	24	107	18.3%	**3.27**	**1.83**	**5.85**
60–69 years	82	93	46.9%	39	140	21.8%	**3.17**	**1.99**	**5.03**
70+ years	25	97	20.5%	12	78	13.3%	1.68	0.79	3.55
Place of living									
Countryside	73	77	48.7%	30	81	27.0%	**2.56**	**1.51**	**4.34**
Small-sized town (<50 thousand)	35	39	47.3%	21	67	23.9%	**2.86**	**1.47**	**5.59**
Medium-sized town (51–100 thousand)	39	89	30.5%	20	147	12.0%	**3.22**	**1.77**	**5.87**
Large-sized town (>100 thousand)	19	94	16.8%	10	89	10.1%	1.80	0.79	4.08
Marital status									
Married or cohabiting	132	194	40.5%	62	305	16.9%	**3.35**	**2.36**	**4.76**
Widow	18	57	24.0%	11	34	24.4%	0.98	0.41	2.31
Never married	0	31	-	0	41	-	-	-	-
Divorced or separated		27	-	1	27	3.6%	-	-	-
Education									
ISCED 5–6	6	88	6.4%	7	143	4.7%	1.39	0.45	4.28
ISCED 3–4	128	185	40.9%	65	232	21.9%	**2.47**	**1.73**	**3.52**
ISCED 1–2	34	36	48.6%	12	33	26.7%	**2.60**	**1.16**	**5.84**
BMI									
<25 kg/m^2^	82	123	40.0%	49	324	13.1%	**4.41**	**2.92**	**6.64**
25–30 kg/m^2^	61	109	35.9%	27	57	32.1%	1.18	0.68	2.06
≥30 kg/m^2^	25	78	24.3%	9	29	23.7%	1.03	0.43	2.47
Age of first menstrual period									
≤12 years	56	132	29.8%	20	93	17.7%	**1.97**	**1.11**	**3.51**
13–15 years	98	141	41.0%	57	245	18.9%	**2.99**	**2.03**	**4.40**
16–18 years	14	36	28.0%	8	72	10.0%	**3.50**	**1.34**	**9.11**
Age of menopause									
≤44 years	6	20	23.1%	8	14	36.4%	0.53	0.15	1.85
45–49 years	9	36	20.0%	11	17	39.3%	0.39	0.13	1.11
50–54 years	72	105	40.7%	37	193	16.1%	**3.58**	**2.25**	**5.68**
≥55 years	50	81	38.2%	12	56	17.6%	**2.88**	**1.41**	**5.90**
Number of pregnancies									
0	13	50	20.6%	3	77	3.8%	**6.67**	**1.81**	**24.61**
1	27	83	24.5%	17	137	11.0%	**2.62**	**1.35**	**5.10**
2	57	121	32.0%	36	133	21.3%	**1.74**	**1.07**	**2.83**
3	53	35	60.2%	21	49	30.0%	**3.53**	**1.82**	**6.88**
≥4	18	21	46.2%	8	14	36.4%	1.50	0.51	4.39
Duration of breastfeeding									
0 months	45	111	28.8%	16	79	16.8%	**2.00**	**1.06**	**3.79**
<6 months	65	91	41.7%	29	82	26.1%	**2.02**	**1.19**	**3.43**
6–12 months	48	71	40.3%	32	189	14.5%	**3.99**	**2.36**	**6.74**
>12 months	0	9	-	5	10	33.3%	-	-	-
Smoking									
Former smoker	56	106	34.6%	19	94	16.8%	**2.61**	**1.45**	**4.71**
Non-smoker (including passive smoker)	45	87	34.1%	32	90	26.2%	1.45	0.85	2.50
Smoker	47	35	57.3%	22	36	37.9%	**2.20**	**1.10**	**4.37**

Source: own calculations. Bold: statistically significant ORs.

**Table 2 ijerph-18-04570-t002:** The crude odds ratios of breast cancer in night workers (taking into account the intensity of night work) compared to non-night workers.

Exposure to Night Shift Work	Case	Control	OR	95% CI
Ever worked night shift work				
Never	310	410	1	Reference
Ever	168	85	**2.61**	**1.94–3.53**
Number of consecutive night shifts				
No night shift work	310	410	1	Reference
≤3 consecutive night shifts	43	28	**2.03**	**1.23–2.10**
>3 consecutive night shifts	114	50	**3.02**	**3.34–4.34**
Direction of the shift rotation				
No night shift work	310	410	1	Reference
Forward rotation	148	76	**2.58**	**1.88–1.03**
Backward rotation	10	4	**3.31**	**3.52–10.64**
Years of night shift work before illness				
No night shift work	310	410	1	Reference
≤10 years night shift work before illness	8	10	1.06	0.41–2.71
>10 years night shift work before illness	154	70	**2.91**	**2.12–4.00**
Number of night work years	310	510	1	Reference
No night shift work				
1–9 years of night work	19	17	1.48	0.76–2.89
10–19 years of night work	**74**	**31**	**3.16**	**2.02–4.92**
20–29 years of night work	**44**	**20**	**2.91**	**1.68–5.04**
30–39 years of night work	**27**	**14**	**2.55**	**1.32–4.95**

Source: own calculations. Bold: statistically significant ORs.

**Table 3 ijerph-18-04570-t003:** Adjusted odds ratios of breast cancer in night workers (taking into account the intensity of night work) in the last row, with the statistically significant ORs for confounding factors in rows above the last one.

Variable\Model Ver. No.	A	B	C	D	E	F	G
BMI > 25	3.43	3.36	3.45	3.38	3.47	3.34	3.42
Breastfeeding < 6 months	2,74	2.71	2.74	2.87	2.71	2.68	2.84
Menstruation age: 10–12	2.15	2.14	2.15	2.17	2.15	2.14	2.16
Menopause age: 55+	1.86	1.81	1.86	1.82	1.80	1.86	1.78
Living in countryside	1.76	1.81	1.80	1.75	1.73	1.86	1.75
Widow	1.62	1.58	1.67	1.68	1.67	1.61	1.70
Smoking	1.57	1.58	1.55	1.62	1.56	1.69	1.62
No pregnancies	0.55	0.54	0.54	0.52	0.54	0.51	0.52
A–F version of night work *	2.24 ^A^	2.21 ^B^	2.10 ^C^	2.44 ^D^	2.40 ^E^	1.031 ^F^	2.66 ^G^

Source: own calculations in Gretl and Statistica 13.3. *****
^A^: any night shift work, ^B^: working nights more than 10 years before illness, ^C^: forward rotation, ^D^: more than three consecutive night shifts, ^E^: more than 10 years of working nights, ^F^: sum of the night shift nights (trend variable), ^G^: more than 10 years of working nights and more than three consecutive night shifts.

## Data Availability

The data presented in this study are available on request from the corresponding author. The data are not publicly available due to privacy restrictions.

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
