# Peer review of "How the Intensity of Night Shift Work Affects Breast Cancer Risk"

_ijerph, 2021, doi:10.3390/ijerph18094570_

Round 1

Reviewer 1 Report

Dear authors,

This is a very interesting article. Thank you for giving me the opportunity to read and review this manuscript. I think that your article will provide a relevant study about the topic. However, I suggest the following suggestion, in order to be finally accepted:

Although the paper cites 32 references, It could go a little deeper into the analysis of the state of the art, especially with the incorporation of more recent studies on the subject.

Author Response

Thank you very much for your review and comments. Please see the attachment,

Kind regards,

Marta Szkiela

Reviewer 2 Report

This article is for association between intensity of night shift work and breast cancer. The authors mentioned their previous study (ref 25) very often. The readers could understand this article after reading previous article.

1. The authors should delete the description of previous article especially in results section.

2. In method section, table, and result section, a detailed description of the variables included in the logistic regression analysis should be added. 

3. The table expression is too difficult to understand. A detailed description should be added to make the table itself to make it easier to understand. 

For example,

1) Table 1: OR- what OR? OR or adjusted OR? which variables are adjusted?

Confounding factors should be described at footnote.

 What is important, night work proportion within each case/ control group or night work/non night work proportion in case group?

If the former is more important, column should be categorized as following

Case Control
Night workers Non Night workers Night workers Non Night workers
N % N N % N

 % : percent at each row within each case/control group.

Table name should be included the meaning of OR. 

Table1. The odds ratios of night work compared to non night work at each category.

Table 2 and 3 should be renamed.

Table 3 is very difficult to understand. Almost all information should be included within table. Is A-G at first row different with last row?

4. The statistical method should be described in detail to help readers understand the results.

5. At line 156 and 157, 495 and 478 must swap positions with each other.

6. Please see the following reference.

Night Work and Breast Cancer Risk in Nurses: Multifactorial Risk Analysis.

Gómez-Salgado J, Fagundo-Rivera J, Ortega-Moreno M, Allande-Cussó R, Ayuso-Murillo D, Ruiz-Frutos C.Cancers (Basel). 2021 Mar 23;13(6):1470. doi: 10.3390/cancers13061470.    

Author Response

Thank you very much for the generally positive opinion about the scope of our study. Relevant corrections have been made according to your recommendations. 

Please see the attachment,

Kind regards,

Marta Szkiela

Reviewer 3 Report

Manuscript ID: ijerph-1176855 

The work is devoted to an important topic - to study the effect of shift work on the risk of developing breast cancer. It should be noted that there are currently quite a large number of publications devoted to this topic. The advantage of this work is that in their analysis they took into account some ontogenetic factors (breastfeeding) and the state of reproductive function (the beginning of puberty, number of pregnancies, the age of menopause), which also have a significant impact on the risk of breast cancer. A significant drawback, from my point of view, is that the authors did not take into account in their study another factor directly related to the topic under study: the state of the circadian system. It is known that people have a different ability to adapt to night shifts, depending on their chronotype, which has a significant impact on the risk of cancer. Taking this factor into account would allow the authors to provide more constructive recommendations for the prevention of breast cancer in shift workers. Below are some comments on some sections of the manuscript

Introduction

  1. P. 2 Paragraph 2. It is also necessary to indicate the role of different chronotypes and their ability to adapt to night shifts in the risk of cancer.

Methods

  1. It is necessary to indicate the method by which the anthropometric data of the research participants were assessed. If it is a self-assessment method, then at the end of the discussion section it is necessary to indicate this as a limitation of the study.
  2. In Statistical methods section it is necessary to indicate the estimates goodness of fit of the models.

Discussion

  1. In the Discussion section, the authors did not discuss the chronobiological aspects of the results obtained, although in the Introduction they indicated that desynchronosis is the main mechanism for increasing the risk of breast cancer in shift workers.
  2. It is necessary to cite the classic works of Schernhammer E., devoted to the problem considered in the manuscript.
  3. At the end of the Discussion section, it is necessary to indicate the limitations of the study.

Conclusion

  1. In the Conclusion section, it is necessary to briefly indicate the main results of the study without references to tables and literature sources.

Author Response

(The authors gave the same response as above.)

Round 2

Reviewer 2 Report

In method section, please describe the method and purpose of analysis in table 3.

Table 3 should be added reference at each variable column. For example, BMI>25 (ref. BMI<=25). (A)~(G) should be marked with superscript without (). At first row, It is better to denote A~G as model 1~7 and please add the explanation for each model at footnote.

"culumnt" in Line 164-165 should be corrected.

Author Response

Thank you very much for your review. Please see the attachment.

Kind regards,

Marta Szkiela

Reviewer 3 Report

In the Limitation  section, the authors need to specify that the anthropometric indicators were evaluated by the self-assessment method.

Author Response

Thank you very much for your review. Please see the attachment,

Kind regards,

Marta Szkiela
